# Differential linear brain growth patterns in preterm neonates based on birth gestational age and steroid exposure: A retrospective chart review

Medha Goyal[1,2*], Meagan Quigley[3], Sourabh Dutta[4], Nina Stein[5], Ipsita Goswami[2]

1 Clinical Fellow (Neonatal-Perinatal Medicine Program), Division of Neonatology, Department of Pediatrics, University of Toronto, Ontario, Canada, 2 Division of Neonatology, Department of Pediatrics, McMaster Children's Hospital, McMaster University, Ontario, Canada, 3 Faculty of Health Sciences, McMaster University, Hamilton, Ontario, Canada, 4 Division of Neonatology, Department of Pediatrics, Postgraduate Institute of Medical Education and Research (PGIMER), Chandigarh, India, 5 Department of Medical Imaging, McMaster Children's Hospital, McMaster University, Hamilton, Ontario, Canada,

* medha.goyal@mail.utoronto.ca, goyalm5@mcmaster.ca

## Abstract

### Objective

To assess the differences in brain growth between extreme preterm [EP](22–28wks gestation age [GA]) and very preterm infants [VP](28[+1]–32wks GA) using two-dimensional cranial ultrasound(cUS) at term equivalence.

### Study design

Retrospective study of neonates born at GA of ≤ 32 weeks between 1st January 2019 and 31st December 2022, without major parenchymal brain injury.

### Results

326 neonates, with 207 EP and 119 VP, were enrolled. EP infants compared to VP had significantly lower biparietal diameter [7.7vs7.9 cm, p = 0.003], corpus-callosum length [3.8vs4.1 cm, p < 0.001], corpus-callosum-fastigial distance [4.5vs4.8 cm, p = 0.004] and cerebellar-vermis height [2.1vs2.2 cm, p = 0.002]. Cumulative postnatal steroid exposure had no significant association with brain metrics; however, exposure to antenatal steroids was negatively associated with corpus-callosum length [β = −0.38 (−0.58 to −0.7),p = 0.0003] and pons anteroposterior depth [β = −0.36 (−0.47 to −0.25),p < 0.0001] despite adjustments for clinically important risk factors.

### Conclusion

Preterm infants born ≤ 28 weeks GA have significantly smaller dimensions of major white matter tracts than preterm infants born 28–32 weeks GA at term equivalence.

**Data availability statement:** All relevant data are within the paper and its Supporting Information files.

**Funding:** The author(s) received no specific funding for this work.

**Competing interests:** The authors have declared that no competing interests exist.

**Abbreviations:** BPD, Bronchopulmonary dysplasia; BW, Birthweight; CGA, Corrected gestation age; CSF, Cerebrospinal fluid; cUS, Cranial ultrasound; EP, Extreme preterm; GA: Gestational age; hsPDA, Hemodynamically significant patent ductus arteriosus; IQR, Interquartile range; IVH, Intraventricular hae-morrhage; MRI, Magnetic resonance imaging; NEC, Necrotising enterocolitis; NICU, Neonatal Intensive Care Unit; PHVD, Post-hemorrhagic ventricular dilatation; RDS, Respiratory distress syndrome; ROP, Retinopathy of prematurity; SGA, Small-for-gestational-age; VP, Very preterm.

Exposure to antenatal steroids negatively impacts corpus-callosum length and pons anteroposterior depth.

---

## Introduction

Advancement of perinatal and postnatal care practices over the last decade has led to a dramatic improvement in the survival rates of preterm infants born as early as 22 weeks gestational age (GA). [1] Preterm birth is associated with significant risks of brain injury and impaired brain development. [2] Preterm brain injury increases the likelihood of cerebral palsy [3], cognitive impairment and behavioural concerns [4], and poor academic achievements, all of which adversely affect the quality of life and lifelong earning capacity. All preterm infants ≤32 weeks routinely undergo cranial ultrasounds (cUS) serially from birth to term equivalence (36–40 weeks GA) to detect signs of brain injury and sequelae. [5] Diagnosis of Grade III-IV intraventricular hemorrhage (IVH), periventricular leukomalacia (PVL) and post-hemorrhagic ventricular dilation (PHVD) predict significant neurodevelopmental disabilities. [6] However, many preterm infants will have either no signs of brain injury or Grade I-II IVH on routine cUS. [7] Irrespective of the degree of early brain injury, term-equivalent imaging is highly recommended as a prognostic tool for future neurodevelopmental outcomes, especially in high-risk preterm infants ≤29 weeks GA. [8] Although magnetic resonance imaging (MRI) is superior to cUS in detecting white matter injury [5], cUS is the most feasible imaging modality in most neonatal units worldwide.

Preterm brain growth trajectory can be altered even in the absence of severe brain injury. Infants born <32 weeks GA without parenchymal brain injury exhibited impaired global and regional brain growth in the cerebrum, cerebellum, and brainstem detected by MRI compared with the in-utero healthy fetuses. [9] Potential clinical factors contributing to altered brain development include environmental stressors [10], ventilation strategies or corticosteroid exposure [11], suboptimal nutrition [12], intermittent hypoxia [13], and infection/inflammation. [14] The impact of antenatal corticosteroid exposure has been shown to be variable, with negligible change in regional brain volumes in neonates born at 23–32 weeks GA at term equivalence. [15]. Another retrospective cohort reported reduced amygdala, caudate nucleus, and hippocampus volumes in neonates between 28 and 34 weeks at term equivalence, with no impact in those < 28 weeks. [16] We need to elucidate the impact of corticosteroids in larger subsets to determine the differences in brain growth in neonates born at < 28 weeks and at > 28 weeks GA.

Preterm birth may interrupt the stochastic developmental processes of the fetal brain.[17] Prenatal brain development occurs through neuronal migration, differentiation and synaptogenesis, myelination, and cortical folding.[18] Since different brain regions and functions develop at different gestations, the timing of preterm birth is likely to have differential disruptive effects on brain structures. Previous studies have used cUS to compare brain dimensions in neonates with different GA. Biparietal diameter and basal ganglia width were significantly smaller in neonates with birth GA of 32–33 weeks and 34–35 weeks GA infants than full-term infants.[19] Moreover,

there is an association between cUS-based linear metrics and long-term neurodevelopment outcomes in these infants. Greater corpus callosum fastigial length at 2 months corrected age in preterm infants was associated with higher cognitive score, while corpus callosal length was positively associated with cognitive, motor and language outcomes at 2 years.[20] The cUS-based linear brain growth metrics of preterm infants born ≤ 32 weeks GA have not been widely researched.

This study aimed to (i) assess the differences in specific brain regions between neonates born extreme preterm [EP (22–28 weeks GA)] and very preterm [VP (28+1–32 weeks GA)] without major parenchymal brain injury using two-dimensional cUS and (ii) study the effect of antenatal and postnatal corticosteroid exposure on linear brain metrics in neonates born ≤32 weeks GA, adjusting for relevant clinical risk factors.

## Methodology

A retrospective cohort study was conducted at the tertiary care neonatal unit at McMaster Children's Hospital, Ontario, Canada. The Institutional Ethics Board approved the study (#16327) and data was accessed on 15/06/2023. All consecutive preterm neonates with a birth GA of ≤ 32 weeks admitted between 1st January 2019 and 31st December 2022 were eligible to be enrolled in the study. Neonates with (i) grade III IVH, (ii) periventricular hemorrhagic infarct, (iii) PVL, (iv) PHVD, (v) major congenital cranial malformations; (vi) major chromosomal abnormalities or aneuploidies were excluded. Any neonate with non-availability of cUS at term equivalent GA was excluded.

Neonates delivered at 22–28 weeks GA (EP) were compared with those delivered at 28+1–32 weeks GA (VP). Maternal and neonatal characteristics were collected from electronic health records. Common neonatal morbidities during neonatal intensive care unit (NICU) stay were recorded, including respiratory distress syndrome (RDS), grade I-II IVH, retinopathy of prematurity (ROP), necrotizing enterocolitis (NEC), culture-positive sepsis, hemodynamically significant patent ductus arteriosus (hsPDA) and bronchopulmonary dysplasia (BPD). We did not exclude neonates with the diagnosis of meningitis or suspected meningitis due to concerns with data validity due to wide variation in the interpretation of CSF results and lack of universal criteria for diagnosis. We recorded the exposure to antenatal and postnatal systemic corticosteroids. Antenatally, mothers received either none or one dose or two doses of betamethasone. Postnatally, neonates were exposed to either no systemic steroids or systemic steroids in the form of hydrocortisone and/or dexamethasone. The cumulative dose of postnatal glucocorticoid exposure was calculated as prednisolone-equivalent dose (g/kg); i.e., Dexamethasone = (5 × total dose per prescription)/0.75 and Hydrocortisone (5 × total dose per prescription)/20. [21] We did not include inhaled steroids such as budesonide as the systemic absorption can be challenging to quantify accurately.

## Neuroimaging

The cUS performed between 36–40 weeks corrected GA was retrospectively reviewed by one investigator (MG) in conjunction with a neuroradiologist (NS). When multiple cUS were done between 36–40 weeks corrected GA, only the first cUS was used for data analysis. The cUS images were obtained by trained ultrasound technicians using the Canon Aplio i-series 700 machine [Canon Medical Systems Corporation, Japan] with 7 megahertz and 18 megahertz curvilinear probes for superficial and deeper structures, respectively. All cUS were done following the standard neuroimaging protocol for the unit. Coronal views were obtained from the anterior fontanelle window by sweeping the entire brain from anterior to posterior. Sequential sagittal views were recorded, assessing structures on the right, midline, and left sides. Cerebellar structures were visualized through the mastoid view from both sides.

Images recorded were reviewed in six coronal and five sagittal planes using the anterior fontanelle window and at least one coronal and one axial plane using the mastoid fontanelle window.[22] The following parameters were measured from digitally archived images representative of different brain regions, namely (i) Biparietal diameter, (ii) Cerebral White Matter [Corpus-callosum length and Corpus-callosum-fastigial distance], (iii) Deep Gray Matter [Basal ganglia width, caudate head width], (iv) Brain Stem [Pons anteroposterior depth], and (v) Cerebellum [Cerebellar vermis height and

Trans-cerebellar diameter.[23] The measurements were obtained in mid-coronal, parasagittal, and mid-sagittal planes. We calculated an average of the two sides for parameters with measurements in the right and left cerebral hemispheres. The detailed descriptions of various measurements are described in S2 File.

## Statistical analysis

All statistical analyses were performed using GraphPad Prism Version 10.3.0 (GraphPad Software, LLC., San Diego, CA, USA). Standard descriptive statistics were used to compare the baseline characteristics of the two groups. Linear metrics were compared between the two groups using the Mann-Whitney U test for non-parametric data and the Student T-test for parametric data. To study the effect of sex and intrauterine growth restriction in each group, we performed a two-way ANOVA with Bonferroni's correction for multiple comparisons. Linear regression was used to investigate the relationship between birth GA and cumulative steroid exposure with linear brain metrics. We investigated the association between corticosteroid exposure and brain metrics using multivariate linear regression adjusting for relevant clinical risk factors.

## Results

### Population characteristics

Three hundred twenty-six preterm neonates with a mean GA and birth weight (BW) of $27 \pm 3$ weeks and $1096 \pm 418$ grams, respectively, were enrolled in the study. The study flow diagram is described in S2 File. Among them, 169 (52%) were males, 70 (22%) were twins and 10 (3%) were triplets. Key neonatal morbidities included severe RDS requiring ≥1 dose of surfactant 258 (79%), IVH grade I or II 183 (56%), NEC stage II-III 24 (7%), hsPDA 106 (33%), culture-positive sepsis 90 (28%), BPD 233 (72%), and ROP requiring treatment 19 (6%). Among infants with BPD, the incidence of mild, moderate and severe BPD was 7 (3%), 45 (20%), and 181 (77%), respectively. The median (1st, 3rd quartile) hospital stay was 90 (46, 136) days.

Table 1 compares the maternal and neonatal characteristics of EP (N = 207 infants) and VP (N = 119 infants). The mean (SD) GA of EP and VP were $26 \pm 1.7$ weeks and $30 \pm 1.2$ weeks, respectively. EP neonates had a significantly lower mean birth weight [891 (258) g versus 1453 (405) g, p-value < 0.001] and higher rates of vaginal delivery [75 (36%) versus 27 (22%), p-value 0.011] and singleton pregnancy [163 (79%) versus 83 (70%), p-value < 0.001] compared to VP. EP needed more resuscitation at birth and had a higher incidence of all key neonatal morbidities, along with longer hospital stays (Table 1).

### Comparison of linear brain metrics

The mean (SD) corrected GA when cUS was analyzed in EP and VP were similar at $37 \pm 2$ weeks. EP had significantly lower median (1st, 3rd quartile) biparietal diameter [7.7 (7.2,8.2) cm versus 7.9 (7.4,8.8) cm, p-value = 0.003], corpus callosal length [3.8 (3.5,4.2) cm versus 4.1 (3.8,4.5) cm, p-value = 0.0002], corpus callosum-fastigial distance [4.5 (4.3,4.4) cm versus 4.8 (4.4,5.1) cm, p-value = 0.004] and cerebellar vermis height [2.1 (1.8,2.3) cm versus 2.2 (1.9,2.7) cm, p-value = 0.002] compared to VP (Fig 1). The two groups had no statistically significant difference regarding other brain metrics. (S3 File).

In the entire cohort, neonates with small-for-gestational-age (SGA) < 10th birth centile on Modified Fenton's growth charts (N = 28), had a smaller mean (SD) biparietal diameter than neonates without SGA (N = 298) [7.5 (1.2) cm versus 7.8 (1.1) cm, p-value = 0.017]. Furthermore, neonates with SGA had significantly smaller mean (SD) basal ganglia width compared to neonates without SGA [1.76 (0.37) cm versus 1.89 (0.37) cm, p-value = 0.026]. We further found no significant interaction of SGA between the two GA groups for any of the brain metrics. (S4 File (top panel)). Further, in the entire cohort, female neonates had a significantly smaller median (1st, 3rd quartile) biparietal diameter [7.6 (7.1, 8.2)] cm

**Table 1. Comparison of the maternal and neonatal characteristics of the two groups.**

| | EP infants [22-28 weeks GA] N=207 | VP infants [28⁺¹-32 weeks GA] N=119 | P value |
|---|---|---|---|
| Maternal age (years)[a] | 31.5 (5.5) | 31.9 (5.4) | 0.40 |
| IVF pregnancy n (%) | 20 (9.6) | 11 (9.2) | 0.90 |
| Maternal hypertension n (%) | 38 (18.3) | 22 (18.4) | 0.98 |
| **Maternal diabetes** n (%) | **17 (8.2)** | **26 (21.8)** | **0.0005** |
| Recreational drugs exposure n (%) | 30 (14.4) | 15 (12.6) | 0.63 |
| **Singleton** n (%) | **163 (78.7)** | **83 (69.7)** | **0.0001** |
| Small for gestational age n (%) | 15 (7.2) | 13 (10.9) | 0.25 |
| 2 doses of antenatal steroids n (%) | 21 (10.1) | 10 (8.4) | 0.44 |
| Magnesium Sulphate n (%) | 178 (85.9) | 93 (78.1) | 0.07 |
| **Vaginal delivery** n (%) | **75 (36.2)** | **27 (22.7)** | **0.011** |
| Delayed Cord Clamping n (%) | 125 (60.3) | 71 (59.6) | 0.90 |
| Male sex n (%) | 110 (53.1) | 59 (49.5) | 0.54 |
| **Birth weight** [a] | **891 (258)** | **1453 (405)** | **<0.0001** |
| **Apgar at 5 min** [b] | **7 (6, 8)** | **8 (7, 9)** | **<0.0001** |
| **Respiratory distress syndrome** n (%) | **190 (91.7)** | **68 (57.1)** | **<0.0001** |
| **Grade I-II IVH** n (%) | **132 (63.8)** | **51 (42.9)** | **0.0006** |
| **Necrotizing Enterocolitis** n (%) | **21 (10.1)** | **3 (2.5)** | **0.011** |
| **Hemodynamically significant PDA** n (%) | **98 (47.3)** | **8 (6.7)** | **<0.0001** |
| **Culture proven sepsis** n (%) | **73 (35.2)** | **17 (14.2)** | **<0.0001** |
| **Bronchopulmonary Dysplasia** n (%) | **180 (86.9)** | **53 (44.5)** | **<0.0001** |
| **Length of stay (days)** [b] | **111 (76, 155)** | **39 (12, 89)** | **<0.0001** |

[a]Median (Standard deviation) [b] Median (1st, 3rd quartile); Abbreviations: PDA (patent ductus arteriosus), IVH (intraventricular hemorrhage), IVF (in-vitro fertilization)

compared to males [7.9 (7.4, 8.8)] cm (p=0.0076). Sex did not significantly interact with the GA groups for all brain metrics, except for trans-cerebellar diameter, which was significantly smaller for girls compared to boys in EP only (**Fig 2**, S4 File **(bottom panel)**).

## Association with corticosteroid exposure

Among 326 infants, 182 (56%) received none, 113 (35%) received one dose, and 31(10%) received two or more doses of antenatal corticosteroids before delivery. Postnatally, 138 (42%) infants received either dexamethasone or hydrocortisone, 71(22%) infants received dexamethasone, 42(13%) infants received hydrocortisone, and 25(8%) infants received both dexamethasone and hydrocortisone. The median (1st, 3rd quartile) postnatal age at initiation of postnatal corticosteroid was 23 (12, 37) days of life. The mean (range) of the cumulative corticosteroid dose received was 6.25 (0, 581) g/kg of prednisone equivalent. The maternal and neonatal characteristics of those exposed to antenatal steroids compared to those not exposed to any antenatal steroids are compared in S5 File.

When adjusted for gestational age, sex, receipt of antenatal steroids, BPD, IVH and hsPDA, there was no significant association between exposure to systemic postnatal corticosteroids [β coefficient for postnatal steroids] and the brain metrics, specifically trans-cerebellar diameter [β=0.11 (95% CI −0.11 to 0.34), p=0.31]; biparietal diameter [β=0.14 (95% CI −0.13 to 0.42), p=0.29], corpus-callosum length [β=−0.06 (95% CI −0.20 to 0.07), p=0.36], corpus callosum-fastigial distance [β=0.011 (95% CI −0.12 to 0.14), p=0.86] and cerebellar vermis height [β=−0.03 (95% CI −0.17 to 0.11), p=0.64].

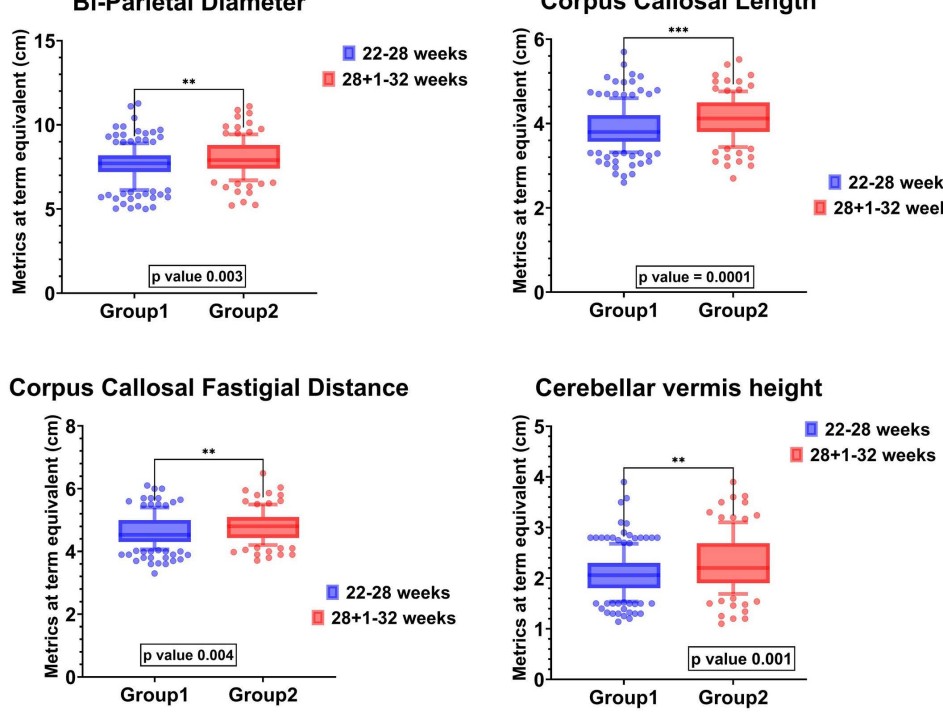

**Fig 1. Brain linear metrics on cranial ultrasound at term equivalence comparing dimensions of biparietal diameter, corpus-callosum length, corpus-callosum-fastigial length and cerebellar vermis height between EP (22-28 weeks GA) and VP (28 +1-32 weeks GA) infants.**

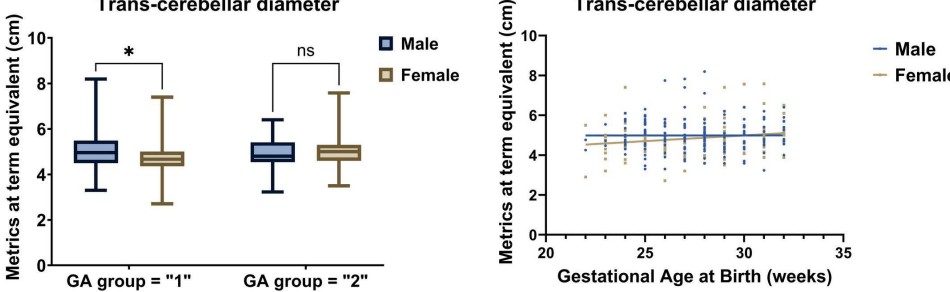

**Fig 2. Linear metrics of transverse cerebellar diameter (TCD) between male and female infants in two groups [*left:* significantly smaller in EP female infants compared to male infants (*p-value* =0.016), *right:* Simple linear regression of TCD in male and female infants based on gestational age].**

There was no statistically significant association between cumulative doses of postnatal corticosteroid exposure and any of the measured brain metrics [**Fig 3**].

For neonates who were exposed to any antenatal steroids, there was a statistically significant negative association despite adjustments [sex, GA, BPD, IVH (Grade 1–2), PDA, sepsis] on the corpus-callosum length and pons anteroposterior depth (**Table 2**). The biparietal diameter, basal ganglia width, and caudate nucleus width were statistically preserved in neonates with antenatal steroid exposure with adjustments for sex, GA, BPD, IVH, PDA and sepsis. (**Table 2**). There

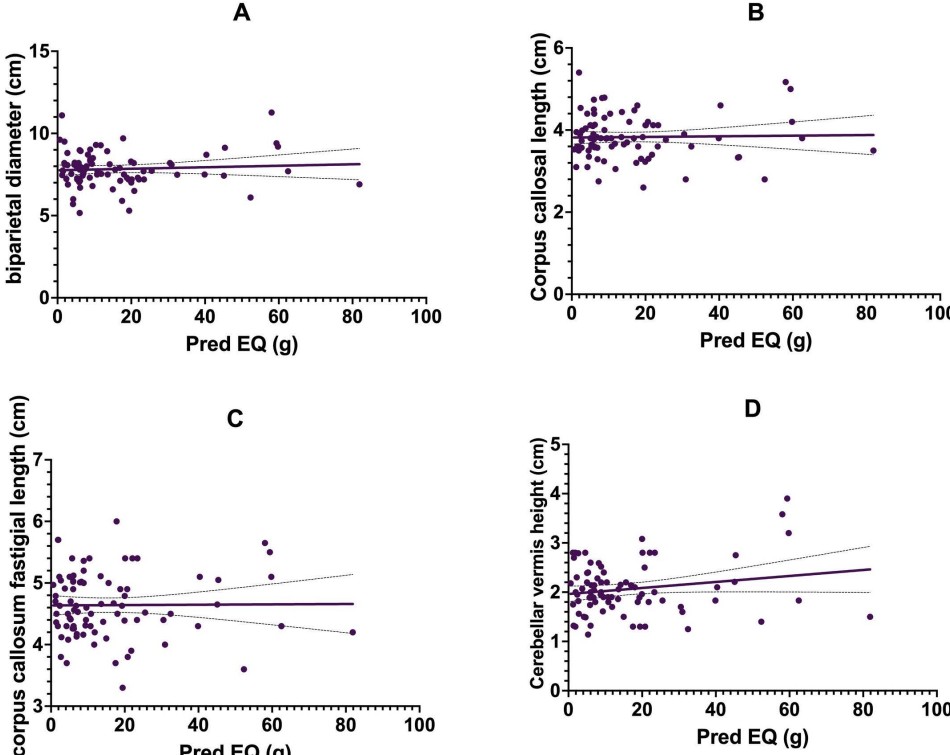

**Fig 3. Linear regression of cumulative postnatal systemic steroid exposure measured in Prednisolone equivalents (EQs) with brain linear metrics of biparietal diameter, corpus-callosum length, corpus-callosum-fastigial length and cerebellar vermis height.**

**Table 2. Comparison of differential brain growth represented by brain linear metrics at term equivalence on cranial ultrasound in neonates with any antenatal corticosteroid exposure versus no exposure.**

| Brain Metric | Antenatal Steroid exposure | Unadjusted β coefficient | | Adjusted* β coefficient | |
|---|---|---|---|---|---|
| | | Estimate (95% CI) | p-value | Estimate (95% CI) | p-value |
| Biparietal Diameter | ≥ 1 dose | 1.13 (0.7 to 1.5) | <0.0001 | 1.11 (0.7 to 1.5) | <0.0001 |
| | No dose | 1.22 (0.8 to 1.6) | <0.0001 | 1.17 (0.8 to 1.6) | <0.0001 |
| Basal ganglia width | ≥ 1 dose | 0.39 (0.24 to 0.54) | <0.0001 | 0.39 (0.23 to 0.54) | <0.0001 |
| | No dose | 0.46 (0.32 to 0.60) | <0.0001 | 0.45 (0.31 to 0.60) | <0.0001 |
| Caudate nucleus head width | ≥ 1 dose | 0.13 (0.05 to 0.21) | 0.0010 | 0.14 (0.06 to 0.22) | 0.0006 |
| | No dose | 0.13 (0.06 to 0.21) | 0.0004 | 0.14 (0.07 to 0.22) | 0.0003 |
| Corpus callosum length | ≥ 1 dose | −0.42 (−0.62 to −0.23) | <0.0001 | −0.38 (−0.58 to −0.7) | 0.0003 |
| | No dose | −0.48 (−0.68 to −0.28) | <0.0001 | −0.34 (−0.54 to −0.15) | 0.0007 |
| Corpus callosum fastigial length | ≥ 1 dose | −0.03 (−0.24 to 0.17) | 0.76 | 0.05 (−0.15 to 0.26) | 0.60 |
| | No dose | −0.05 (−0.25 to 0.14) | 0.61 | 0.02 (−0.17 to 0.22) | 0.82 |
| Cerebellar vermis height | ≥ 1 dose | −0.61 (−0.80 to −0.42) | <0.0001 | 0.13 (−0.07 to 0.34) | 0.20 |
| | No dose | −0.71 (−0.90 to −0.53) | <0.0001 | 0.14 (−0.05 to 0.34) | 0.15 |
| Transverse cerebellar diameter | ≥ 1 dose | 0.13 (−0.19 to 0.45) | 0.43 | 0.19 (−0.15 to 0.52) | 0.27 |
| | No dose | 0.20 (−0.11 to 0.51) | 0.21 | 0.24 (0.08 to 0.56) | 0.14 |
| Pons anteroposterior depth | ≥ 1 dose | −0.38 (−0.49 to −0.28) | <0.0001 | −0.36 (−0.47 to −0.25) | <0.0001 |
| | No dose | −0.37 (−0.47 to −0.26) | <0.0001 | −0.34 (−0.45 to −0.24) | <0.0001 |

*Adjusted for antenatal steroids, sex, gestational age, bronchopulmonary dysplasia, intraventricular hemorrhage (Grade 1–2), hemodynamically significant patent ductus arteriosus, sepsis

were no statistically significant differences in the corpus callosum- fastigial length, cerebellar vermis height, and transverse cerebellar diameter despite antenatal steroid exposure.

## Discussion

This study compares linear metrics of brain regions in EP and VP infants through cUS assessment at term equivalence in infants without significant parenchymal brain injury, except for mild IVH and subependymal bleeds. EP infants had a higher incidence of neonatal morbidities compared to VP infants. EP infants had significantly lower biparietal diameter, corpus callosal length and fastigial distance, and cerebellar vermis height than VP infants; however, they had no differences in basal ganglia or caudate nucleus dimensions. Further, despite the absence of Grade III-IV IVH, PVL, or PHVD, EP infants had smaller white matter structures than VP infants.

Almost 45% of the cohort was exposed to one or two doses of antenatal corticosteroids, and 42% was exposed to postnatal systemic corticosteroids. Complete steroid exposure was seen in nearly 10% of our population, which aligns with the temporal trends of optimal steroid prophylaxis reported between 10–20% over a period of 24 years in Canada. [24]

Neonates exposed to antenatal steroids had smaller linear metrics of corpus callosum length and pons anteroposterior diameter after adjusting for sex, GA, BPD, IVH, hsPDA, and sepsis. Further dimensions of biparietal diameter, basal ganglia and caudate nucleus head width were preserved despite antenatal steroid exposure. This differential impact could stem from the possible slowing of specific critical stages of fetal brain development at which they are administered in the antenatal period. Major neurogenic events, including neuronal proliferation, migration and organization, occur in six transitional compartments in the fetal telencephalon during the second and third trimesters.[25] Proliferative zones, such as intermediate or sub-plate zones, contribute to synaptogenesis from deep-to-superficial pattern peak between 26–32 weeks and help develop the cortical plate.[26] However, the development of white matter tracts, including myelination, continues beyond birth for several months to years.[24] This finding is important to understand the direct impact of antenatal corticosteroids on various stages of telencephalon reorganization in the fetal brain.

Early postnatal brain size measured by cUS has been associated with neonatal neurobehavior [27] and is a marker of neurodevelopmental outcomes. [1] Corpus callosum and corpus callosum–fastigium length have a high intra-class correlation coefficient of > 0.97. [28] There is a good correlation between 2-D cUS measurements in the preterm brain and 3-Tesla MRI-based measurements for biparietal diameters, corpus callosal length, and cerebellar diameter. [29] Haggman *et al*. published the reference values for all linear metrics in full-term neonates without brain injury.[23] Compared to full-term infants, infants born at 32–33 weeks GA and late preterm infants had a smaller biparietal diameter and basal ganglia-insula width but larger corpus callosum fastigium length. [19] In our study, we found differences between the EP and VP infants in the biparietal diameter, corpus callosum and cerebellar vermis, without any difference in deep gray matter structures, potentially indicating a difference in the regional growth of white matter rather than gray matter structures. Wu et al. reported faster growth rates of cerebellar hemispheres and pons than the cerebellar vermis and midbrain/medulla in preterm infants compared to healthy term controls. [30] It is also likely that EP infants are frequently exposed to cumulative effects of intermittent hypoxia, systemic inflammation and fluctuations of cerebral blood flow, which slows white matter growth.

Sex has been reported to have differential effects on brain growth, with male preterm infants having higher intracranial volume and cerebral white matter but proportionally lower cerebral cortical gray matter by 12 months,[31] leading to larger brain volumes with altered microstructure.[32] Although this was observed in full-term neonates, it is less evident in preterm infants < 37 weeks and non-existent in preterm infants < 32 weeks.[33] The weaker associations are believed to be due to confounding by additional neonatal morbidities in preterm infants. We also observed no sex differences in the brain metrics, except for biparietal diameter in the entire cohort and trans-cerebellar diameter in EP neonates. Both metrics were noted to be larger in male infants, conforming to the reports from previous studies. [31–32] The reason why this sex association is non-uniformly distributed across the brain regions could be genetically determined [34] or a methodological variation. Future studies will need to verify the reproducibility of the findings.

Fetal growth restriction has been associated with abnormal fetal brain topology [35], reduced cerebellum and supratentorial volume ratio [36], and decreased gray [35] and white matter. [37] Several mechanisms may lead to neurologic injuries in SGA neonates exposed to growth restriction, such as neuronal apoptosis, neuroinflammation, oxidative stress, injury by excitatory amino acids, disruption of the blood-brain barrier, and epigenetics. [38] Using advanced neuroimaging, growth-restricted infants were noted to have smaller gray matter volumes (thalamus and basal ganglia), whereas there was no difference in cerebellar volumes after adjusting for sex, GA, and weight. [39] Similar to previous reports, our cohort also showed a significantly lower biparietal diameter and basal ganglia width in growth-restricted patients, but there was no significant interaction with the birth GA, indicating that the differences in the gray matter region were perhaps solely influenced by fetal growth restriction.

A significant proportion of infants in the study cohort received systemic postnatal corticosteroids. However, we did not observe any statistically significant effect of exposure to steroids or cumulative doses of steroids received during the hospital stay on any linear brain metrics. The effect of postnatal corticosteroid therapy on the developing brain and the likelihood of cerebral palsy after receiving therapy for BPD is controversial. An earlier meta-analysis reported the typical relative risk for cerebral palsy ranged from 1.66 to 2.86, with a number needed to harm 11. [40] More recently, treatment with dexamethasone for longer than 14 days was associated with lower scores in the motor and language domains compared with unexposed infants. [41] Exposure to dexamethasone was associated with a reduction in volumes across multiple brain regions, including the cerebellum; however, no such association was found for hydrocortisone. [42] The absence of a significant effect of postnatal steroids on linear metrics could be potentially related to the methodological constraints of two-dimensional ultrasound. Firstly, all previous studies have demonstrated differences in volumes of brain structure, which can be measured either by MRI or 3-dimensional ultrasound. [42] Secondly, the effect of postnatal steroids is predominantly microstructural changes such as microglial dysregulation. [43]

This study is limited by its retrospective study design. The use of two-dimension cUS does not allow the determination of brain volumes. Additionally, we might have included a cohort of neonates who were sicker and stayed in Level III NICU till 36 weeks GA and underwent cUS since more stable neonates were transferred to Level II NICUs before 36 weeks GA. We did not include head circumference measurements due to concerns with a lack of standardization with measurement technique and possible large inter-observer variability leading to over or underestimation of head circumferences documented in the records. This limits our ability to correlate our measurements with head circumference. Despite limitations, we present a large sample of extremely preterm infant data that has not been widely reported in the past. Our cohort was sicker neonates who needed Level III NICU admission until term equivalence, which makes the findings more relevant for neonates at the highest risk of poor brain growth.

## Conclusion

When measured using two-dimensional cUS at term equivalence, preterm infants born at or less than 28 weeks GA have significantly smaller linear dimensions of major white matter tracts than preterm infants born between 28- and 32 weeks GA, even in the absence of parenchymal brain injury. This study found no significant association between cumulative doses of postnatal corticosteroids and linear brain metrics of preterm infants born < 32 weeks GA. On the contrary, exposure to antenatal steroids was associated with smaller dimensions of corpus-callosum and pons anteroposterior depth. The long-term functional implications of our findings will need further elucidation.

## Supporting information

**S1 File. Measurements for brain linear metrics performed on cranial ultrasound at term equivalence age along with plane of measurement.**
(DOCX)

**S2 File. Study flow chart.**
(JPG)

**S3 File. Measurements for brain linear metrics on term equivalent cranial ultrasound that were not significantly different between Group 1 (22–28 weeks GA) and Group 2 (28$^{+1}$–32 weeks GA).**
(DOCX)

**S4 File. Top panel: (*Left*) Comparison of linear brain metrics between SGA and non-SGA infants in EP (22–28 weeks GA) and VP (28$^{+1}$–32 weeks GA) at term equivalence age; (*Right*) Simple linear regression of linear brain metrics of SGA and non-SGA infants.** Bottom panel: (*Left*) Comparison of linear brain metrics between male and female EP (22–28 weeks GA) and VP (28$^{+1}$–32 weeks GA) infants at term equivalence age; (*Right*) Simple linear regression of linear brain metrics of male and female infants.
(JPG)

**S5 File. Comparison of maternal and neonatal characteristics of neonates with steroid and without steroid exposure.**
(DOCX)

**S6 File. STROBE checklist.**
(DOCX)

## Author contributions

**Conceptualization:** Medha Goyal, Nina Stein, Ipsita Goswami.

**Data curation:** Medha Goyal, Meagan Quigley.

**Formal analysis:** Medha Goyal, Meagan Quigley, Sourabh Dutta, Nina Stein, Ipsita Goswami.

**Investigation:** Medha Goyal, Meagan Quigley, Sourabh Dutta, Nina Stein, Ipsita Goswami.

**Methodology:** Medha Goyal, Meagan Quigley, Sourabh Dutta, Nina Stein, Ipsita Goswami.

**Project administration:** Medha Goyal, Sourabh Dutta, Nina Stein, Ipsita Goswami.

**Resources:** Medha Goyal, Nina Stein, Ipsita Goswami.

**Software:** Medha Goyal, Sourabh Dutta, Nina Stein, Ipsita Goswami.

**Supervision:** Sourabh Dutta, Nina Stein, Ipsita Goswami.

**Validation:** Sourabh Dutta, Nina Stein, Ipsita Goswami.

**Visualization:** Medha Goyal, Ipsita Goswami.

**Writing – original draft:** Medha Goyal, Meagan Quigley.

**Writing – review & editing:** Medha Goyal, Sourabh Dutta, Nina Stein, Ipsita Goswami.

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
