## [Decision Letter · Decision Letter 0]

18 Feb 2025

PONE-D-25-01288Differential linear brain growth patterns in preterm neonates based on birth gestational age and steroid exposure: A retrospective chart reviewPLOS ONE

Dear Dr. Goyal,

Thank you for submitting your manuscript to PLOS ONE. After careful consideration, we feel that it has merit but does not fully meet PLOS ONE’s publication criteria as it currently stands. Therefore, we invite you to submit a revised version of the manuscript that addresses the points raised during the review process.

We look forward to receiving your revised manuscript.

Kind regards,

Kazumichi Fujioka

Academic Editor

PLOS ONE

Reviewers' comments:

Reviewer's Responses to Questions

**Comments to the Author**

1. Is the manuscript technically sound, and do the data support the conclusions?

Reviewer #1: Yes

Reviewer #2: Partly

2. Has the statistical analysis been performed appropriately and rigorously? 

Reviewer #1: Yes

Reviewer #2: I Don't Know

3. Have the authors made all data underlying the findings in their manuscript fully available?

Reviewer #1: Yes

Reviewer #2: No

4. Is the manuscript presented in an intelligible fashion and written in standard English?

Reviewer #1: Yes

Reviewer #2: No

5. Review Comments to the Author

Reviewer #1: This is an extremely instructive research on the differences in brain growth between extremely preterm infants and very preterm infants. I will only point out the issues that might be problematic, in my honest opinion.

- Line 47 – please, add ”at term equivalence”, either at the end of the conclusion or at the beginning

- Line 99 and lines 297-298 – moderately preterm infants have a GA of 32-33 weeks and late preterm infants have GAs of 34-36 weeks. This is also the way the cited resource uses the classification

- Line 100 – please add ”in these infants” to the next sentence (”Moreover, in these infants there is an association […]”)

- Line 173 – I believe the authors mean ”≥” instead of ”>”, otherwise I wonder where are the infants needing only one dose of surfactant – not including those seem wrong on many levels

- For Table 1 and the Figures – I recommend the authors stop using Group 1 and Group 2 and keep using EP(I) and VP(I), it becomes confusing which is which

- Table 1 and lines 201-207 – I strongly recommend the authors use SGA instead of IUGR, as the definition they use is that of SGA. Later on, in the Discussions section (lines 327-338), they may admit to using the term SGA as a proxy for fetal/intrauterine growth restriction

- Just out of personal curiosity: is the reason listed in lines 361-364 also valid for the small number of VP infants? I was intrigued when first reading the manuscript by the number of VPI, which is smaller that the number of EPI, it is usually the other way round... or is it another reason for the disproportionate groups? I think the authors should at least attempt to explain this

- In Figures 1 and 3 – please, correct ”Corpus callosum”

Reviewer #2: I appreciate the opportunity to review this interesting manuscript on brain growth between EP and VP using a large study population. However, it seems to have several significant problems as below.

Major points

1. Abstract: The conclusions didn’t include the effect of antenatal steroids, but the title included that. It is better to be unified.

2. Introduction: The rationale for the aims of this study should be clarified by citing adequate papers. The effect of antenatal steroids on brain growth in preterm infants has already been reported in a few studies; doi: 10.1016/j.ejogrb.2024.08.034., doi: 10.1542/peds.2004-0326.etc. Then, please describe any new findings that the authors have tried to add.

2. Study design: The authors should show the flowchart of the study population. Were the placental abruption included in this study?

3. Table 1: Information including histological amnionitis and funisitis also have some effect on neonatal brain growth. This information is necessary to understand the results of this study. Please show them.

4. Table 1: If two doses of antenatal steroids mean a complete course of antenatal steroids, the population seems too low (10.1% vs. 8.4 %). If it is correct, is the low prevalence of antenatal steroids common in Canada? Please describe and discuss it.

5. Results. (L225): What does ‘two or more antenatal corticosteroids’ mean? Two doses are a single complete course. If multiple courses are included, those populations should be divided into groups from a complete course of antenatal steroids.

6. Results. The authors also should show the Maternal and neonatal characteristics regarding the comparison shown in Table 2.

7. Results. (L244-251): It is confusing. Did the authors compare the data between single

255 versus no exposure, which is shown in Table 2. The sentences described the comparison between none or one dose of steroids versus a complete course of antenatal steroids.

8. Results. The population of a complete course of antenatal steroids (n=31) is too small to adjust for seven factors, including antenatal steroids, sex, gestational age, bronchopulmonary dysplasia, intraventricular hemorrhage(Grade 1-2), hemodynamically significant patent ductus arteriosus, sepsis. Please consult a statistician. Then the authors should discuss the limitations of this study.

9. Discussion. It is too long. Please focus on the results of this study and explain straightforwardly.

6. PLOS authors have the option to publish the peer review history of their article (what does this mean? ). If published, this will include your full peer review and any attached files.

**Do you want your identity to be public for this peer review?** For information about this choice, including consent withdrawal, please see our Privacy Policy .

Reviewer #1: **Yes: ** Andreea AVASILOAIEI

Reviewer #2: No

---

## [Author Response · Author response to Decision Letter 1]

4 Mar 2025

Reviewer #1: This is an extremely instructive research on the differences in brain growth between extremely preterm infants and very preterm infants. I will only point out the issues that might be problematic, in my honest opinion.

- Line 47 – please, add ”at term equivalence”, either at the end of the conclusion or at the beginning

We have added “term equivalence” at the end of the conclusion.

- Line 99 and lines 297-298 – moderately preterm infants have a GA of 32-33 weeks and late preterm infants have GAs of 34-36 weeks. This is also the way the cited resource uses the classification

We have changed the GA to reflect moderate preterm infants as 32-33 weeks.

- Line 100 – please add ”in these infants” to the next sentence (”Moreover, in these infants there is an association […]”)

We have added the phrase to the sentence.

- Line 173 – I believe the authors mean ”≥” instead of ”>”, otherwise I wonder where are the infants needing only one dose of surfactant – not including those seem wrong on many levels.

We apologize for the overlook and have changed it to ≥.

- For Table 1 and the Figures – I recommend the authors stop using Group 1 and Group 2 and keep using EP(I) and VP(I), it becomes confusing which is which.

To improve clarity in Table 1, we have changed the terminology from Groups 1 and 2 to EP and VP. In the figures, we have additional data for gestational ages with colour coding, which helps readers understand.

- Table 1 and lines 201-207 – I strongly recommend the authors use SGA instead of IUGR, as the definition they use is that of SGA. Later on, in the Discussions section (lines 327-338), they may admit to using the term SGA as a proxy for fetal/intrauterine growth restriction

We have changed the term to SGA.

- Just out of personal curiosity: is the reason listed in lines 361-364 also valid for the small number of VP infants? I was intrigued when first reading the manuscript by the number of VPI, which is smaller that the number of EPI, it is usually the other way round... or is it another reason for the disproportionate groups? I think the authors should at least attempt to explain this

As we have mentioned in lines 363-364, the disproportion was predominantly due to most VP infants doing well and transferring to level 2 centres in the community. Hence, their cranial ultrasounds were not recorded, and they were not included.

- In Figures 1 and 3 – please, correct ”Corpus callosum”

As both terms – “corpus callosum” and “corpus callosal,” are used interchangeably in the literature, we chose to use either in our paper.

Reviewer #2: I appreciate the opportunity to review this interesting manuscript on brain growth between EP and VP using a large study population. However, it seems to have several significant problems as below.

Major points

1. Abstract: The conclusions didn’t include the effect of antenatal steroids, but the title included that. It is better to be unified.

We have added the effect of antenatal steroids in the conclusion as well.

2. Introduction: The rationale for the aims of this study should be clarified by citing adequate papers. The effect of antenatal steroids on brain growth in preterm infants has already been reported in a few studies; doi: 10.1016/j.ejogrb.2024.08.034., doi: 10.1542/peds.2004-0326.etc. Then, please describe any new findings that the authors have tried to add.

We have added these two references and elucidated the gap in our current knowledge regarding the impact of antenatal corticosteroids on regional areas of the preterm developing brain. [Lines 94-100]

2. Study design: The authors should show the flowchart of the study population. Were the placental abruption included in this study?

We have added the flowchart of the study population. Placental abruption was not included in the study.

3. Table 1: Information including histological amnionitis and funisitis also have some effect on neonatal brain growth. This information is necessary to understand the results of this study. Please show them.

Unfortunately, we did not include this information in our research protocol, so we do not have this data.

4. Table 1: If two doses of antenatal steroids mean a complete course of antenatal steroids, the population seems too low (10.1% vs. 8.4 %). If it is correct, is the low prevalence of antenatal steroids common in Canada? Please describe and discuss it.

We have described the prevalence of antenatal steroid use in Canada, and the trends point towards a low prevalence of complete steroid coverage. [Lines 363 – 365]

5. Results. (L225): What does ‘two or more antenatal corticosteroids’ mean? Two doses are a single complete course. If multiple courses are included, those populations should be divided into groups from a complete course of antenatal steroids.

31 (10%) received two or more antenatal corticosteroids before delivery. None received more than two doses. Since the maximum doses received were two, we have clarified the statement.

6. Results. The authors also should show the Maternal and neonatal characteristics regarding the comparison shown in Table 2.

In Supplementary File 4, we have described antenatal steroid exposure in detail and compared maternal and neonatal characteristics.

7. Results. (L244-251): It is confusing. Did the authors compare the data between single versus no exposure, which is shown in Table 2. The sentences described the comparison between none or one dose of steroids versus a complete course of antenatal steroids.

We have clarified the statements and apologize for the lack of clarity. We compared any antenatal steroid exposure to no antenatal steroid exposure and have updated both the text and table to improve the clarity of the results.

8. Results. The population of a complete course of antenatal steroids (n=31) is too small to adjust for seven factors, including antenatal steroids, sex, gestational age, bronchopulmonary dysplasia, intraventricular hemorrhage(Grade 1-2), hemodynamically significant patent ductus arteriosus, sepsis. Please consult a statistician. Then the authors should discuss the limitations of this study.

We have compared the neonates who received no antenatal steroids with those who received ≥ 1 dose of antenatal steroids. We apologize for the lack of clarity in the table, leading to confusion. The group with no antenatal steroid exposure = 182, and for those who had ≥ 1 antenatal steroid exposure = 144, hence we adjusted for seven factors. We have one of the co-authors trained in biostatistics who reviewed the results.

9. Discussion. It is too long. Please focus on the results of this study and explain straightforwardly.

We have improved the discussion section by focusing on study results and re-framing it to make it more comprehensive and clear to readers.

---

## [Decision Letter · Decision Letter 1]

9 Apr 2025

Differential linear brain growth patterns in preterm neonates based on birth gestational age and steroid exposure: A retrospective chart review

PONE-D-25-01288R1

Dear Dr. Goyal,

We’re pleased to inform you that your manuscript has been judged scientifically suitable for publication and will be formally accepted for publication once it meets all outstanding technical requirements.

Kind regards,

Kazumichi Fujioka

Academic Editor

PLOS ONE

Additional Editor Comments (optional):

Reviewers' comments:

Reviewer's Responses to Questions

**Comments to the Author**

1. If the authors have adequately addressed your comments raised in a previous round of review and you feel that this manuscript is now acceptable for publication, you may indicate that here to bypass the “Comments to the Author” section, enter your conflict of interest statement in the “Confidential to Editor” section, and submit your "Accept" recommendation.

Reviewer #1: All comments have been addressed

Reviewer #2: All comments have been addressed

2. Is the manuscript technically sound, and do the data support the conclusions?

Reviewer #1: Yes

Reviewer #2: Yes

3. Has the statistical analysis been performed appropriately and rigorously? 

Reviewer #1: Yes

Reviewer #2: I Don't Know

4. Have the authors made all data underlying the findings in their manuscript fully available?

Reviewer #1: Yes

Reviewer #2: Yes

5. Is the manuscript presented in an intelligible fashion and written in standard English?

Reviewer #1: Yes

Reviewer #2: Yes

6. Review Comments to the Author

Reviewer #1: All of my comments have been addressed. I believe the manuscript is suited to be published in the present form.

Reviewer #2: I appreciate your great effort to answer my questions. All concernes have been adequately addressed.

7. PLOS authors have the option to publish the peer review history of their article (what does this mean? ). If published, this will include your full peer review and any attached files.

**Do you want your identity to be public for this peer review?** For information about this choice, including consent withdrawal, please see our Privacy Policy .

Reviewer #1: No

Reviewer #2: No

---

## [Editor Report · Acceptance letter]

PONE-D-25-01288R1

PLOS ONE

Dear Dr. Goyal,

I'm pleased to inform you that your manuscript has been deemed suitable for publication in PLOS ONE. Congratulations! Your manuscript is now being handed over to our production team.

Kind regards,

on behalf of

Dr. Kazumichi Fujioka

Academic Editor

PLOS ONE